# Harvester Productivity in Inclined Terrain with Extended Machine Operating Trail Intervals: A German Case Study Comparison of Standing and Bunched Trees

**Ferréol Berendt [1,2,*]**, **Eduardo Tolosana [3]**, **Stephan Hoffmann [1]**, **Paula Alonso [1,3]** **and Janine Schweier [1,4]**

[1]  Chair of Forest Operations, Faculty of Environment and Natural Resources, University of Freiburg, 79085 Freiburg, Germany; stephan.hoffmann@canterbury.ac.nz (S.H.); paula.alonso.rod@gmail.com (P.A.); janine.schweier@wsl.ch (J.S.)

[2]  Department of Forest Utilization and Wood Markets, Faculty of Forest and Environment, Eberswalde University for Sustainable Development, 16230 Eberswalde, Germany

[3]  E.T.S.I. de Montes, Forestal y del Medio Natural, Universidad Politécnica de Madrid, 28040 Madrid, Spain; eduardo.tolosana@upm.es

[4]  Research Group Sustainable Forestry, Swiss Federal Institute for Forest, Snow and Landscape Research WSL, 8903 Birmensdorf, Switzerland

*  Correspondence: ferreol.berendt@hnee.de

**Abstract:** The complexity of highly structured forests with multiple tree species, especially when coniferous and broadleaved tree species are mixed, as well as stands with extended machine operating trail spacing and inclined terrain, create challenging operational conditions for mechanized timber harvesting and extraction. Motor-manually felling trees within the midfield and bunching them at the machine operating trails, prior to the arrival of a harvester-forwarder system, is a complex operation. The aim of this study was to assess and compare tethered harvester productivities of a thinning operation, for felling and processing standing trees and for processing bunched trees, through a time study in forest stands with 40-m distances between machine operating trails. Total operational costs of the analyzed thinning operation were 69 €/$m^3_{o.b.}$, including extraction using a multiple forwarder approach. Tree species, merchantable timber volume, and whether the trees were standing or presented as bunched logs all had a significant effect on the harvester time consumption. Moreover, harvester positioning time was significantly shorter when trees were already bunched at the machine operating trail. While the productivity of standing or bunched spruce trees did not differ significantly between the cases (approximately 18 $m^3_{o.b.}$/productive machine hours excluding all delays ($PMH_0$)), the productivity of standing broadleaved tree species (8.3 $m^3_{o.b.}$/$PMH_0$) was much lower than that of bunched trees (15.5 $m^3_{o.b.}$/$PMH_0$). Thus, the described timber harvesting and extraction system may be a valuable option for forest stands with high proportion of broadleaved trees.

**Keywords:** forest operations; machine operating trail; midfield; single-grip harvester; soil protection; tethering winch

## 1. Introduction

Through high structural and species diversity within managed forests, the resistance and resilience of the forests towards future climatic conditions can be improved [1]. However, such management approaches will lead to more complex forest stands, and thus the complexity and potential impacts caused by timber harvesting operations might increase as well [2]. Tree species and diameter, terrain

factors such as slope, soil strength, and soil roughness, and operational factors, such as the forest road system, restrict in particular fully mechanized wood harvesting and extraction operations [3–5]. Timber extraction is usually conducted by using ground-based systems in flat and intermediate terrain and cable-based systems in steep terrain. However, depending on the wood species, amount, and assortments, the latter system might be a challenge in terms of economic viability [6,7]. At the same time, modern systems using tethering winches are becoming increasingly common and are recommended to be used in flat and intermediate terrain, but few studies were conducted on productivity, cost-effectiveness, environmental or work safety impacts, in order to fully rate these systems [7].

Forest operations are subject to technical limitations and personal constraints of the forest owners, associated with the individual management objective, but they also have to fulfil legal regulations and, if accredited by a forest certification scheme, the corresponding standard. Examples of international forest certification schemes are the Programme for the Endorsement of Forest Certification (PEFC) and the Forest Stewardship Council (FSC). The latter determines, for the German standard, that less than 13.5% of the forest area should be used as machine operating trails or forest roads, due to soil protection reasons [8]. The resulting 40-m machine operating trail spacing is also encouraged by many state forest enterprises, such as ForstBW of Baden-Württemberg [9]. Inevitably, such regulations lead to a machine operating trail system with midfields that are outside the boom reach of the harvester and where trees need to be felled motor-manually. Subsequently, full trees are bucked at the machine operating trail through a harvester, or—if too short—full trees or assortments are pre-winched to the machine operating trails using, for example, mini forestry crawlers (MFC) or tractors with a winch, and bunched for further processing [10,11].

The aim of this study is to gain knowledge on the application of timber harvesting and extraction systems with midfields, which are used for pre-winching and bunching trees. We hypothesized that the harvester productivity of bunched trees is higher than that of standing trees in the boom reach of the harvester. Thus, the specific goals of this study were:

1.  to analyze the time consumption of ground-based wood harvesting and extraction on inclined terrain with midfields;
2.  to compare the harvester productivity of standing and bunched trees;
3.  to deduce further research goals from the study outcomes.

## 2. Materials and Methods

### 2.1. Study Approach and Site

We collected data as part of a case study in the German federal state of Baden-Württemberg (BW), more precisely in the municipality of Kleines Wiesental in the Black Forest mountains. The observed operation was a commercial thinning in terrain with slopes from 30% up to approximately 60% and was conducted by a private forestry contractor in October and November 2018. Forest operations in this kind of terrain are conventionally classified for cable yarder systems in Germany [12]. However, the forest manager did not consider using a cable yarder, due to (i) technical infeasibility and (ii) expected higher costs. Further, (iii) work safety was an important aspect because many trees were broken by snow.

To avoid long skidding distances and the resulting considerable stand damage, a harvester-forwarder system with a tethering winch was applied. Despite being at its upper slope limit of operation, the engaged machines were fully stable at the site while in a stationary position, as well as when the winch-cable was not tensioned, thus meeting requirements for safe operation of traction-assisted equipment by international standards [13]. Nevertheless, for both soil protection and societal acceptance reasons, the machine operating trail intervals were 40 m. Besides the harvesting and forwarding, motor-manual felling and pre-winching to the machine operating trail in the midfield were carried out by the forest company.

The study site, with the coordinates 47°47′19.2″ N and 7°46′58.4″ E, was located between 1100 and 1200 m above sea level. The slope inclination of the machine operating trails ranged between 60% and 70% and was estimated using a clinometer (PM-5, Suunto, Vantaa, Finland). The stand size was 9.9 ha and the dominant tree species was Norway spruce (*Picea abies* H. Karst), with a share of 65%. In addition, the mixed stand consisted of other conifers (16% Douglas fir (*Pseudotsuga menziesii* Franco) and 5% Scots pine (*Pinus sylvestris* L.)), as well as 14% broadleaved species (7% Sycamore maple (*Acer Pseudoplatanus* L.) and 7% European beech (*Fagus sylvatica* L.)). The mean diameter at breast height for these groups was measured prior to the harvesting and reached on average 27.2 ± 5.3 cm for Norway spruce, 33.4 ± 8.4 cm for Douglas fir, and 18.5 ± 10.6 cm for broadleaved tree species. The age of the two main tree species was on average 56 years for Norway spruce and 35 years for Douglas fir. The scheduled harvesting volumes of merchantable timber were 60 to 70 m$^3$ per ha. According to the billings of the forestry company, 670 m$^3_{o.b.}$ were harvested and processed.

## 2.2. Applied System

The distance between machine operating trails was 40 m and the overall operation was conducted as follows, with all working steps carried out one after the other: (1) Trees were felled motor-manually in the midfield using a chainsaw. Moreover, prior to mechanized felling and processing, some trees were felled motor-manually by a second chainsaw operator in order to facilitate the harvester's pass on the machine operating trail. (2) Full trees were partially bunched and pre-winched to the machine operating trails using a remote controlled MFC (Raup-Trac from Martin Alther, Switzerland) that was operated from the machine operating trails. This operation was carried out by a professional two-person team: one felled the trees using the chainsaw while the second operated the MFC. (3) A harvester (405FH4 8WD from HSM, Germany with H415 harvester head from Waratah, New Zealand and Finland) with a tethering winch first felled and processed the standing trees within the boom reach and then processed the bunched trees from the midfields. (4) A forwarder (208F from HSM, Germany) with a tethering winch extracted the processed logs from the machine operating trails to the next forest road, where a second forwarder (HSM 208F Big Foot from HSM, Germany) brought the logs to a roadside landing. The two units were necessary, as the first forwarder was supported by a tethering winch and thus could not leave the straight machine operating trails. Besides extracting timber to the landing site, the second forwarder acted when needed as anchorage for the tethering winch of the first forwarder. This was the case when no tree could be used as an anchor. The representation of the wood harvesting and extraction system according to the German Center for Forest Work and Technology (KWF) [14] is shown in Figure 1.

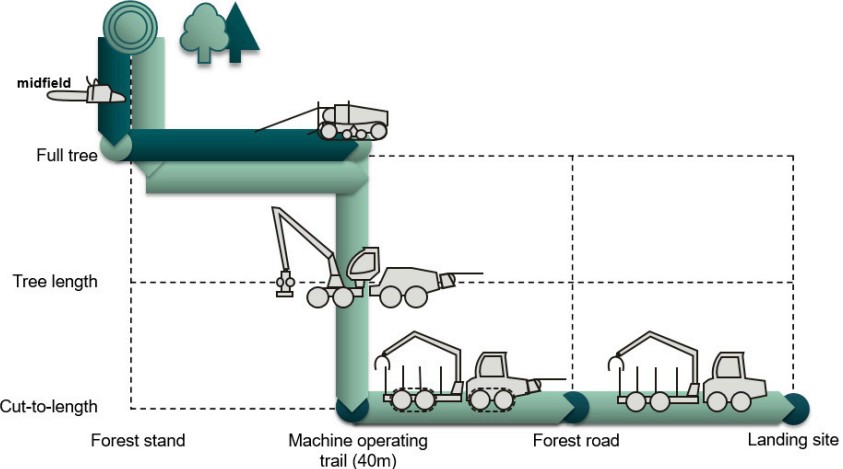

**Figure 1.** Representation of the wood harvesting and extraction operation.

### 2.3. Data Collection

The data collection concentrated on the pre-winching and harvesting operation (steps 2 and 3). An elemental time and motion study, applying the snapback timing approach [15], was conducted with an accuracy of 1/100 min through a digital stop watch in order to monitor the operations of both the MFC and harvester. All delays caused by the study were removed; delays for the lunch break (if any) and relocation to and from site were also not included in the data sets.

The work cycles of the MFC were divided into the following work elements: positioning on the machine operating trails, pulling, chocking, winching, unchocking, and waiting for the feller (including communication between the MFC operator and the feller). Moreover, mechanical delays as well as rest and personal times were recorded.

During data collection concerning the harvester, the cycles were divided into two mandatory work elements that occurred in each cycle: (a) positioning the harvester on the machine operating trail, moving the boom into the direction of the tree, and grabbing the tree ('positioning'); and (b) felling, delimbing and bucking, and stacking the crown ('felling & processing'). These elements correspond to the main work time and complementary work time according to the IUFRO (International Union of Forest Research Organizations) Forest Work Study Nomenclature [15]. Moreover, rest and personal times, as well as supportive work times, were recorded. Within the supportive work times, ancillary work times and time needed for rigging/unrigging the tethering winch cable were recorded. Along with time consumption, the merchantable wood volume in $m^3$ over bark ($m^3_{o.b.}$) was documented through the harvester measurements and on-board computers' records. The harvester operator separated the trees into Norway spruce, Douglas fir, and broadleaved tree species, with the corresponding assortments. Even though Scots pine was observed during the inventory prior to the harvesting operation, Scots pine was not harvested during the thinning. Thus, Scots pine was not considered further in the analysis.

Data concerning the other working steps (1 and 4) were not considered within the time study, since these processes are well studied and overall time consumption and corresponding costs were obtained from the forestry company.

### 2.4. Working Productivity and Costs

A regression analysis was applied in order to determine the effects of merchantable volume, tree species, and the presentation of the logs (i.e., whether the harvester operated with standing or bunched trees on the time consumption). The significance level ($\alpha$) was set to 5%, with $\alpha \leq 1\%$ considered highly significant.

Overall productivity was calculated as the total volume of merchantable timber in $m^3$ over bark ($m^3_{o.b.}$), divided by the total time consumption, whereas the cycle productivity was calculated as the processed timber during one cycle divided by the cycle time consumption. The relationship between harvester productivity and wood volume was fitted through non-linear regression methods in which broadleaved species were excluded, as *n* number of broadleaved tree species was too low. Moreover, data with studentized residues greater than 3.0 were removed as outliers. These were mostly positive residues from cycles longer than normal.

Accurate accounting, with the number of working hours and costs for the whole area (9.9 ha), were obtained from billings of the forestry company and included both, fixed and variable costs. Thus, both machine and operator costs of all processes could be included. The costs were compartmentalized into chainsaw work, MFC, harvester, forwarder 1, forwarder 2, and machine transport from forestry company office to forest area.

## 3. Results

### 3.1. MFC and Harvester Time Consumption

The time consumption for the thinning operations was determined for each working step individually.

Considering the MFC, 178 winching cycles were observed with a mean load volume of 0.49 m$^3$. The recorded scheduled machine hours (SMHs) were 362 min, thereof 286 min referred to productive machine hours (PMH$_0$), resulting in a machine utilization rate (ratio of PMH$_0$ to SMH) of 79%. The overall gross productivity, based on SMHs, was 11.3 m$^3$/SMH, while net productivity was 14.3 m$^3$/PMH$_0$. Within the net cycle time, the longest working elements were winching (25.4%), positioning (24.9%), waiting for the feller (19.7%), and pulling (10.5%). Chocking time corresponded to 9.3% and included rechocking the cable if it was not properly fixed the first time.

For the harvester, total SMHs recorded were 1217 min. Within the 821 cycles, a total of 278.3 m$^3$$_{o.b.}$ was felled, processed, and bunched, corresponding to an overall gross productivity of 13.7 m$^3$$_{o.b.}$/SMH. When looking at the net cycle time, 912.4 min were used as productive time (Table 1). Thus, the machine utilization rate was 75%. The overall time consumption was divided into felling and processing, with 0.65 min per cycle (43.7%), followed by positioning (32.2%), supportive work time (14.8%), and rest and personal time (9.3%). Looking more specifically into the supportive work time, 49.5% was service time (mostly maintenance), 34.5% was ancillary work time (machine operating trails), and 15.9% was preparatory time (setting up and taking down the tethering winch). The mean time consumption for one whole harvester cycle was 1.12 ± 0.78 min. Considering the two work elements separately, mean time consumption was 0.47 ± 0.53 for 'positioning' and 0.65 ± 0.41 min for 'felling & processing'.

**Table 1.** Characteristics of the harvesting operations.

| Characteristic | Bunched Trees | Standing Trees | Sum |
|---|---|---|---|
| Total productive machine hours (PMH$_0$, min) | 496.7 | 415.6 | 912.4 |
| Work cycles | 469 | 352 | 821 |
| Total volume (m$^3$$_{o.b.}$) | 153.0 | 125.1 | 278.1 |
| Mean volume per cycle (m$^3$$_{o.b.}$ ± SD) | 0.33 ± 0.23 | 0.36 ± 0.21 | 0.34 ± 0.23 |
| Number of spruces (pieces) | 273 | 275 | 548 |
| Number of Douglas firs (pieces) | 108 | 50 | 158 |
| Number of broadleaved trees (pieces) | 88 | 27 | 115 |
| Productivity (m$^3$$_{o.b.}$/PMH$_0$) | 18.5 | 18.1 | 18.3 |

### 3.2. Influence of Log Presentation

The merchantable volume differed between species. The average volume of bunched broadleaved trees (0.24 ± 0.29 m$^3$$_{o.b.}$) was larger than the average volume of standing broadleaved trees (0.14 ± 0.09 m$^3$$_{o.b.}$; Figure 2). On the contrary, the average volume of conifers bunched next to the machine operating trails (0.45 ± 0.27 m$^3$$_{o.b.}$ for Douglas fir and 0.30 ± 0.18 m$^3$$_{o.b.}$ for spruce) was smaller than the average volume of standing coniferous trees (0.54 ± 0.33 m$^3$$_{o.b.}$ and 0.34 ± 0.16 m$^3$$_{o.b.}$ for Douglas fir and spruce, respectively). The lowest mean time consumption of one harvest cycle was 0.95 ± 0.63 min for bunched broadleaved trees and the highest mean time consumption was 1.24 ± 0.84 min for standing spruce trees (Figure 3).

Tree volume, tree species, and whether the trees were standing or bunched significantly affected the time consumption of a harvester cycle. Moreover, the tree species had a significant effect on both work elements ('positioning' and 'felling & processing'). When considering the work element 'positioning', the independent variable whether trees were standing or bunched had a highly significant effect; whereas it did not significantly affect the time consumption of the work element 'felling & processing'. On the opposite, the tree species did not affect the time consumption of the work element 'positioning' but did significantly influence the 'felling and processing' time.

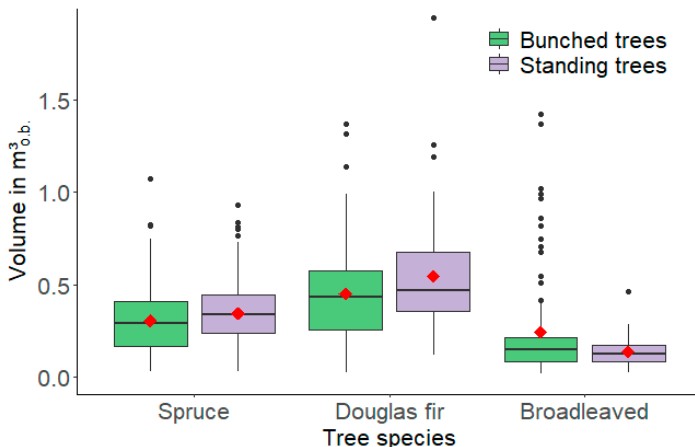

**Figure 2.** Boxplot of the merchantable volume in $m^3_{o.b.}$ for bunched (green) and standing (purple) trees of different species (spruce, Douglas fir and broadleaved tree species) during harvesting operation.

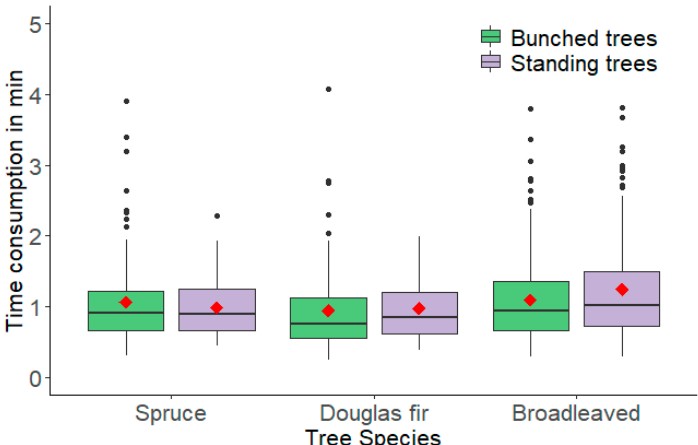

**Figure 3.** Boxplot of the time consumption (in min) for processing standing (purple) and bunched (green) trees of different species (spruce, Douglas fir and broadleaved tree species).

### 3.3. Harvester Productivity

Considering the productive working time, the overall productivity of the harvester was 18.3 $m^3_{o.b.}$/$PMH_0$ (Table 1). The overall net productivity of bunched trees (18.5 $m^3_{o.b.}$/$PMH_0$) was slightly higher than that of standing trees (18.1 $m^3_{o.b.}$/$PMH_0$). However, the tree species distribution was not similar; for example, the share of spruce was 58.2% for the bunched trees and 78.1% for the standing trees. The tree species, as well as the volumes, had a highly significant influence on the cycle productivity, whereas log presentation did not show any significant effect. The highest overall productivity was reached when harvesting Douglas fir, with 27.7 $m^3_{o.b.}$/$PMH_0$, followed by spruce with 16.5 $m^3_{o.b.}$/$PMH_0$, and broadleaved tree species with 13.8 $m^3_{o.b.}$/$PMH_0$. When distinguishing between standing and bunched trees, overall productivity was 46% lower for standing than for bunched broadleaved trees but 30% higher for standing Douglas fir. The harvesting cycle productivity of spruce was not significantly affected by whether the trees were standing or bunched at the machine operating trails, as shown in Figure 4 for all harvesting cycles.

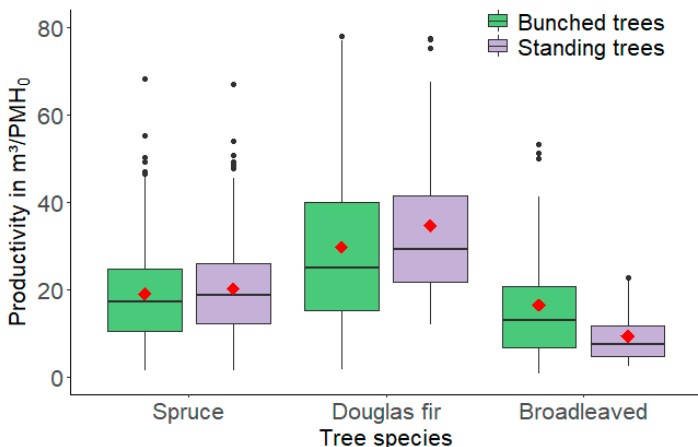

**Figure 4.** Productivity boxplot of the harvesting cycles in $m^3_{o.b.}$ per $PMH_0$ for bunched (green) and standing (purple) trees of different tree species (spruce, Douglas fir and broadleaved tree species) during harvesting operation.

The effect of piece volume on the harvester productivity was analyzed in more detail for coniferous tree species. After removing broadleaved tree species, Equation (1) was fitted using a nonlinear regression technique between productivity and merchantable volume. The expected increase of productivity with increasing volume is shown in Figure 5. While a cycle productivity of 20 $m^3/PMH_0$ was reached for piece volumes of 0.3 $m^3_{o.b.}$, the productivity was expected to be doubled (40 $m^3/PMH_0$) for piece volumes of approximately 1 $m^3$. The coefficient of determination $R^2$ (adjusted by degrees of freedom) was 51.5% (Figure 5). This means that more than half of the variation in harvester cycle productivity was explained by tree merchantable volume.

$$y = 42.51 \times V^{0.637} \tag{1}$$

where y is productivity in $m^3_{o.b.}$ per $PMH_0$ and V is merchantable volume in $m^3_{o.b.}$.

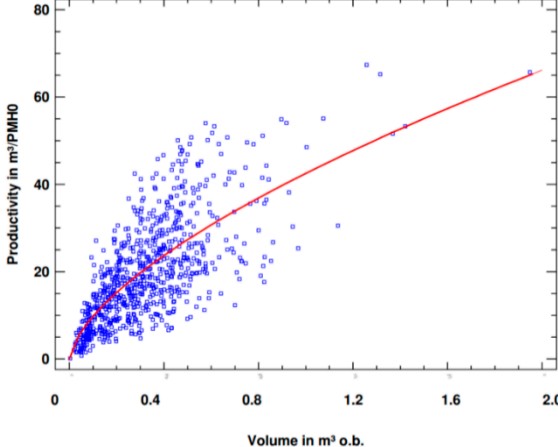

**Figure 5.** Harvester cycle productivity ($m^3_{o.b.}/PMH_0$) of conifers in relation to volume ($m^3_{o.b.}$).

### 3.4. Resulting Costs

The timber harvesting and extraction operation costs totaled 46,323 € or 69 €/$m^3_{o.b.}$ for all four working steps and involved operational units (Table 2). Costs for bunching the trees at the machine operating trail were 18% of the total costs and included the motor-manual felling and winching with the MFC.

**Table 2.** Time consumption and costs of the timber harvesting and extraction operations.

| Operational Units | Hours (Scheduled Machine Hours (SMH)) | Hourly Costs (€/SMH) | Total Costs (€) |
|---|---|---|---|
| Motor-manual felling | 156 | 38 | 5928 |
| Mini forestry crawler | 35 | 70 | 2450 |
| Harvester | 92 | 200 | 18,400 |
| Forwarder 1 | 90 | 90 | 8550 |
| Forwarder 2 | 90 | 93 | 8835 |
| Machine transport | 18 | 120 | 2160 |

## 4. Discussion

Forest managers are currently facing new challenges: (i) changes in forest composition through both an increased number of species and a greater proportion of site-adapted broadleaved species, along with higher structural diversity in order to improve the forest's resistance and resilience towards climate change [1,2]; (ii) changes in forest certification or state regulations on the machine operating trail spacing to support intensified soil protection strategies [8,16]; and (iii) the mechanization of forest operations in inclined terrain, in order to reduce costs and increase worker safety [7]. In this study, we analyzed the performance of a timber harvesting and extraction system, addressing all the above-mentioned challenges. While the additional time consumption for the use of a tethering winch only represented 2.4% of a SMH, it should be noted that one forwarder was used for approximately 28 h as an anchor, contributing a non-negligible cost factor of 5.4% of the total costs.

Compared with a conventional fully mechanized system with 20-m spacing between machine operating trails, the system studied in this case study may involve higher costs, as 18% of the total costs were attributed to motor-manual felling and winching the trees from the midfields to the machine operating trails. Increased costs due to midfield operations were also found in other studies [11,17]. It should be noticed that highly structured forest stands with admixture of broadleaved tree species may lead to processing difficulties when fully mechanized systems are used, which will result in lower productivity and higher costs. Nevertheless, time consumption of MFCs may be improved by better work organization in order to reduce the waiting time (which accounted for nearly 20% of all work time), and thereby reduce the costs. However, increased machine operating trail spacing has positive aspects as well and is in compliance with sustainable forest operations [18]: lower traffic, increased forest production area, and better societal acceptance. Thus, the extra cost for the provision of ecosystem services can be valued as that due to the application of environmental protection regulations and societal acceptance.

In the current study, we used merchantable volumes (not full tree volumes) to quantify production performance. Thus, large differences in the volumes of the different tree species occurred. For a 56-year-old stand, spruce volumes were relatively low and of low quality, due to high rates of tree breakages from snow. On the contrary, due to pruning and early release of future crop trees, Douglas firs had—despite their younger age (35 years)—larger volumes and higher log qualities compared with the Norway spruces. Higher log quality and generally larger log size meant Douglas fir had the highest productivity values, followed by Norway spruce and broadleaved tree species. The latter are known to often complicate and prolong the processing phase due to more complex and variable stems and crown architectures [19].

Processing productivity by the harvester increased significantly for broadleaved tree species when trees were bunched at the machine operating trails. In contrast, considerable positive effects of bunching were not observed for Norway spruce or Douglas fir. Thus, such a system may be more valuable in forest stands with a large proportion of broadleaved tree species. In conifer-dominated stands, other timber harvesting and extraction methods could be more productive; for example, systems using harvesters with a 15-m boom reach. Another system could involve motor-manual felling of the trees in the midfield and winching them with an integrated winch attached to the harvesters.

However, in that case, the harvester operator would need to pay particular attention to worker safety, as forest workers and harvesters would work next to each other. Further, this would lead to potential waiting times until the trees were felled, which could hamper the productivity and thus the efficient use of the harvester. In this respect, a positive aspect of the described timber harvesting and extraction system with midfields is that the motor-manual work, including the bunching and the mechanized harvester-forwarder system, are decoupled in time. This allows more planning and logistical flexibility among the machine systems and additionally increases the safety situation.

It is important to consider that the tethered equipment was at its upper limit for traction-assistance in this study. Depending on regional regulations, work systems similar to the one described here can also be implemented with a winch-assisted set-up. There, the working equipment is indeed fully secured by the cable through an anchor or other machine unit, providing access to terrain the machine would not be able to reach on its own [20]. In terms of worker safety, it is therefore essential to respect the defined operational limits for the type of equipment applied in the designated work system.

## 5. Conclusions

Due to new regulations, an increased share of mixed-species forests and extended machine operating trail spacing are becoming more and more popular in southwestern Germany. The described timber harvesting and extraction system, with its good performance among broadleaved tree species, has proven its suitability for these changing operational conditions. Nevertheless, further field work and research is required to further adapt and improve upon the overall sustainability of such systems, with special attention towards occupational safety and cost reduction. In this respect, a focus should be placed on optimal log presentation towards the harvester's boom reach, at reduced winching distances and motor-manual work effort, with the aim of defining optimal operational layouts, to realize comparable harvester performance as when bunching directly at the machine operating trail.

**Author Contributions:** F.B. conceived and designed the study, with contributions from J.S.; F.B., P.A. and J.S. collected the data; F.B. and E.T. analyzed the data, with contributions from P.A.; F.B., E.T., S.H. and J.S. wrote the manuscript. All authors have read and agreed to the published version of the manuscript.

**Funding:** This study was undertaken within the framework of the project "SOLVE" (Timber harvesting and transportation systems adapted to altered forest structures due to climate change), which was funded by the German Federal Ministry of Food and Agriculture (BMEL) and the German Federal Ministry for the Environment, Nature Conservation, Building and Nuclear Safety (BMUB) in the context of the "Förderrichtlinie Waldklimafond" (Förderkennzeichen 28W-B-3-048-01). J.S. was supported by the European Social Fund and by the Ministry of Science, Research and Arts Baden-Württemberg while working at the University of Freiburg.

**Acknowledgments:** The authors thank Max Bierer, Christian Ludowicy and Mathieu Fortin for their support, as well as the ForstBW A.ö.R., in particular the Maschinenbetrieb Schrofel, the Forstrevier Kleines Wiesental, and the KELLER—Forstbetrieb.

**Conflicts of Interest:** The authors declare no conflict of interest.

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
