# Peer review of "Harvester Productivity in Inclined Terrain with Extended Machine Operating Trail Intervals: A German Case Study Comparison of Standing and Bunched Trees"

_sustainability, doi:10.3390/su12219168_

Round 1
Reviewer 1 Report
The study deals with an actual problem and gives some information on productivity and costs of a special working system, which is applicable under typical conditions not only in BW, but also in hilly terrain anywhere. The case study follows scientific and mathematical standards. The presentation of results, however, is a bit short and could be extended; readers could be interested in more details about time relations, influence of piece volume etc. These informations are important in order to compare the results with other case studies.
Author Response
We wish to thank the reviewer for the constructive comments! Based on your suggestions, we have edited the paper accordingly and included them in the revised manuscript. We have marked changes using the Track Changes Word Tool.
We are sorry not being precise enough in the description of our results. We have revised the section in order to highlight the effect of piece volumes on the productivity. Moreover, we payed attention to describe clearly the independent variables which have a significant effect on the cycle time consumption.
Reviewer 2 Report
In my opinion, the article is very interesting. The aim of the study was to assess and compare tethered harvester productivities, for felling and processing standing trees and for processing bunched trees, through a time study in forest stands with 40-m distances between machine operating trails. Authors analyzed multiple factors significantly influencing costs and productivity, to determine optimum solutions for obtaining the greatest possible volume of timber at the lowest possible cost—optimum in that they represent a compromise between labor input and profit. This method is very commonly used for the estimation of cost, productivity, and other factors for the optimization of forest operations. Considering the diversity of available technologies and the variability of natural and economic conditions, many models serve as a typical case study, and this is signaled.
The manuscript has a typical IMRAD structure. The goal of the paper is clear. The title is clear and reflects the study content. Materials, methods, and results are presented understandably. Materials and methods are described very well. The discussion is consistent with the results.
Other remarks and comments:
Abstract. I suggest add information about stand age or thinning.
L79: Can you approximate what was the proportion of broken trees? I assume that broken trees were also harvested (standing and bunched trees). Are you sure that time consumption was not for this reason higher?
L140-141: Wasn't Scots pine harvested? The species can be omitted from the statistical calculations. However, the costs were calculated on the basis of data obtained from the company! Additional explanations are needed.
L187: Is standard deviation value correct?
L194-199: This part of the text is not clear. Maybe you add the results of regression analysis.
L230: Figure 5 showed results with or without broadleaved tree species? Additional information is needed.
L235 (in table): According to methods: chainsaw (1 person) and mini forestry crawler (1 person)?
L248-251: Confusing; recommend rewriting the sentences. In this case, we have costs with 40-m spacing between operating machine trials. What costs should we take into account for 20-m spacing between operating machine trials (time consumption for harvester and forwarder)? Will the analyzed solution really involve higher costs?
L290-296: The conclusions are general and are not based on the research results.
Concluding, I recommend publishing the paper after successful minor revisions.
Author Response
We wish to thank the reviewer for the constructive comments! Based on your suggestions, we have edited the paper accordingly and included them in the revised manuscript. We have marked changes using the Track Changes Word Tool.
Abstract: Thanks for the comment. We have added in Line 21 that the aim of this study was “to assess […] harvester productivities during a thinning operation”.
L 79: We contacted the forester in charge of the thinning operation to confirm the amount of broken trees by snow. The proportion of broken spruce were above 50 %, while for the other tree species snow breaks was very low. Yes, time consumption may be influenced negatively by the snow breaks. Therefore, we discussed it between line We have discussed it between line 267 and 275.
L.141: To avoid confusion we have revised the sentence. The reason why Scots pine were not further analyzed is that Scots pine were not harvested during the thinning operation. Doing an inventory prior to the thinning operations, we found the proportion of Scots pine was approximately 5 % in the forest stand. Out of the 821 observed harvesting cycles, the logs from Scots pine were negligible. We have added that the proportion was below 0.5 % of the harvesting cycles. We have reformulated the sentence as follow: “Even if Scots pine was observed during the inventory prior to the harvesting operation, Scots pine was not harvested during the thinning. Thus, Scots pine was not considered further in the analysis.”
L.187: Yes, the SD of the mean volume per cycle is correct. It shows that the timber from forest stands was quite heterogeneous. For bunched broadleaved tree species, for example, the high SD could be explained by the many outliers. Moreover, in Figure 2 the heterogeneity of the volumes was represented in a boxplot.
L.194-199: We are sorry not being clear and we revised the paragraph as follow: “Tree volume, tree species and whether the trees were standing or bunched affected significantly the time consumption of a harvester cycle. Moreover, the tree species had a significant effect on both work elements (‘positioning’ and ‘felling & processing’). When considering the work element ‘positioning’, the independent variable whether trees were standing or bunched had a highly significant; whereas it didn’t affect significantly the time consumption of the work element ‘felling & processing’. On the opposite, the tree species did not affect the time consumption of the work element ‘positioning’ but did significantly influence the ‘felling and processing’ time.”
L.230: Thanks, we have added additional information in the caption of Figure 5 and in the text.
L.235: A second chainsaw operator worked only the first day of the thinning operation in order to clean the machine operating trail. We have added that information in the material and methods section as follow: “Moreover, prior to mechanized felling and processing, some trees were felled motor-manually by a second chainsaw operator in order to facilitate harvester’s pass on the machine operating trail”
L.248-251: We have revised the sentences to make it more clear for the readers.
Conslusions: We have changed the conclusion accordingly. Now, the conclusions integrate the research results of our study.